# Gain Adaptation in Sliding Mode Control Using Model Predictive Control and Disturbance Compensation with Application to Actuators

**Benedikt Haus** [1,*] **, Paolo Mercorelli** [1] **and Harald Aschemann** [2]

1   Institute of Product and Process Innovation, Leuphana University of Lüneburg, Volgershall 1,
    D-21339 Lüneburg, Germany; mercorelli@leuphana.de
2   Chair of Mechatronics, University of Rostock, Justus-von-Liebig-Weg 6, D-18059 Rostock, Germany;
    harald.aschemann@uni-rostock.de
*   Correspondence: haus@leuphana.de

**Abstract:** In this contribution, a gain adaptation for sliding mode control (SMC) is proposed that uses both linear model predictive control (LMPC) and an estimator-based disturbance compensation. Its application is demonstrated with an electromagnetic actuator. The SMC is based on a second-order model of the electric actuator, a direct current (DC) drive, where the current dynamics and the dynamics of the motor angular velocity are addressed. The error dynamics of the SMC are stabilized by a moving horizon MPC and a Kalman filter (KF) that estimates a lumped disturbance variable. In the application under consideration, this lumped disturbance variable accounts for nonlinear friction as well as model uncertainty. Simulation results point out the benefits regarding a reduction of chattering and a high control accuracy.

**Keywords:** sliding mode control; model predictive control; adaptive control; disturbance estimation; actuators

---

## 1. Introduction and Literature Review

### 1.1. Adaptive Sliding Mode Control and Model Predictive Control

While belonging to perhaps the most robust and versatile control strategies, sliding mode control (SMC) tends to suffer from high energy consumption and high-frequency oscillations in system inputs, states or even outputs, which certainly is to be avoided in tracking problems. Currently, many remedies have been proposed and successfully realized to deal with these problems. A very important one is the so-called boundary layer approach, see e.g., [1], which introduces a permissible region around the sliding surface. This layer is characterized by its thickness, inside which no switching of the control input takes place. Another frequently applied approach to chattering reduction is higher-order SMC [2], which can also be combined with a boundary layer concept [3]. Nevertheless, powerful alternatives exist towards a model-based or signal-based adaptation of the switching amplitudes. A model-based approach is shown in [4], where model reference adaptive control (MRAC) is employed to adjust the SMC gain in an application of a brushless direct current (DC) motor. The signal-based approaches discussed in the literature are generally based on integral-type scheduling rules, typically case distinctions depending on a norm or an absolute value of the sliding surface, see e.g., [5,6]. The contribution at hand looks at an innovative model-based adaptation. Thanks to countermeasures like these against chattering, SMC is suitable to be widely used in industry. In the context of drive applications, for example, it has even been used to *reduce* torque ripple [7].

Model predictive control (MPC) still constitutes a developing research field in the context of machines and drives, though many applications already use such control strategies, e.g., in [8] for a permanent magnet synchronous machine (PMSM) or in [9] and in [10] for a DC drive. The MPC approach takes into account model-based predictions and determines the control inputs by minimizing a cost function. However, compared to classical controllers like proportional-integral-derivative (PID) control, this method from the field of optimal control requires a relatively high modeling accuracy in order to yield acceptable results. SMC, on the other hand, is known for being robust against disturbances, model mismatch and parametric uncertainties.

### 1.2. Estimation and Kalman Filters

Thanks to the high computing power of modern processors, micro-controllers, or even field-programmable gate arrays (FPGAs), it has become possible to deploy intelligent and sophisticated control approaches, e.g., observer-based control, utilizing only a minimal number of sensors, see [11,12]. Contributions like [13–16] reflect the progress in theoretical studies of Kalman filters (KFs), especially concerning robustness and the ability to deal with unknown or inaccessible disturbances or model uncertainty. In many situations, time-varying disturbances like friction effects can be modeled as additional unknown inputs. In [17] for instance, a two-stage KF is implemented to estimate the pressure disturbance inside a cylinder of an internal combustion engine and its effect on the controlled output.

### 1.3. Actuators

As important mechatronic components, electromagnetic actuators are used in many technical applications, in particular in the automotive industry and in industrial production systems. In production systems, they play a key role in motion control and precise positioning. Mechanical, pneumatic or hydraulic components tend to be replaced by electromagnetic actuators due to their high efficiency, excellent dynamic behavior and cleanliness. An important effect that needs to be considered in the mechanical part of actuators is nonlinear friction. An extended survey of friction modeling is given in [18] including a large number of literature references. Recent contributions mark progress in terms of identification of friction phenomena and their compensation [19].

### 1.4. Main Contribution

In this paper, a combination of SMC and linear MPC is proposed to create an adaptive control method. Here, LMPC adapts the switching height of the discontinuous control part and, thereby, reduces the undesired chattering effect. The combination of SMC and LMPC allows for an exploitation of the benefits of both worlds, gaining both robustness and a degree of optimality with regard to the specific MPC cost function—at the cost of a more sophisticated control design as well as an increased implementation and computation effort. The introduction of a cost function and, therefore, of an optimality measure allows the intuitive balancing of the error convergence rate versus chattering amplitude penalties, in order to achieve a reasonable trade-off. Linear MPC is a straight-forward, easily implementable way of minimizing such a cost function. Furthermore, an augmented linear KF is employed with the primary goal to contribute to a fast convergence to the sliding surface, thus to unburden the switching control part of the SMC and to reduce undesired chattering. To achieve this goal, the KF estimates the disturbance (and also the first derivative with regard to time), which is considered as an unknown input, and is subsequently used for a compensation in the SMC law. Also, since the cost function that is minimized by the MPC includes the predicted tracking error stemming from a model-based prediction scheme, an accurate model of the SMC sliding surface dynamics is necessary to allow for an optimally small SMC gain. To ensure that the underlying model assumptions of the MPC design are met, a compensation of lumped disturbances is mandatory. Thanks to the disturbance compensation using the estimates of the KF, an accurate prediction over a finite horizon becomes possible in compliance with the underlying assumptions at the KF design. The disturbance compensation may also be interpreted as a lowering of the necessary minimum SMC gain because

part of the disturbances are compensated for by the KF estimates—and the SMC is disburdened. In order to demonstrate the properties of the proposed control method in a practice-relevant field, this paper considers a DC drive system that is subject to both nonlinear friction and model uncertainty. The nonlinearity is represented by the sum of the Coulomb friction model and a quadratic term depending on the relative velocity.

To conclude, the goal of this contribution, which represents an extension of a conference paper [20], is to conceive an SMC with optimal adaptivity that can be implemented as simply as possible. Its effectiveness is demonstrated in simulations, subject to realistic conditions regarding disturbances and model uncertainty, controlling the angular velocity of a DC drive including a nonlinear friction model.

The paper is structured as follows:

- Section 2 presents the physically-based model of a DC drive that is affected by a nonlinear friction torque and model uncertainty.
- The feedback control design is described in Section 3, where
- Section 3.1 contains details on the employed SMC techniques which involve a combination of a continuous control action and a discontinuous switching part, and where
- in Section 3.2, the height of the switching control action is adapted using MPC techniques to counteract undesired chattering.
- Moreover, an unknown lumped disturbance—accounting for nonlinear friction and model uncertainty in the equation of motion—is estimated in Section 3.3 by a KF. This estimate is employed subsequently in the error dynamics for compensation purposes and, as a result, contributes to the reduction of the necessary switching height determined by MPC.
- The benefits are shown by meaningful simulation results in Section 4.
- Finally, conclusions are given in Section 5.

## 2. System Modelling

The system model is based on physical considerations and involves ordinary differential equations for the armature current $i(t)$ and the motor angular velocity $\omega(t)$

$$\frac{\mathrm{d}i(t)}{\mathrm{d}t} = \frac{1}{L} \left( u(t) - Ri(t) - K_T \omega(t) \right), \tag{1}$$

$$\frac{\mathrm{d}\omega(t)}{\mathrm{d}t} = \frac{1}{J} \left( K_T i(t) - T_r(\omega(t)) \right). \tag{2}$$

Here, a nonlinear friction torque

$$T_r(\omega(t)) = \left( K_f \omega^2(t) + T_{r0} \right) \mathrm{sign}(\omega(t)) \tag{3}$$

is introduced, where $K_f > 0$ denotes a coefficient related to the quadratic term in the motor angular velocity, and $T_{r0}$ characterizes the Coulomb friction part. The nonlinear friction torque $T_r(\omega) = T_r(\omega(t))$ is depicted in Figure 1. This friction model is implemented in a regularized form and used in the simulation studies to represent nonlinear friction.

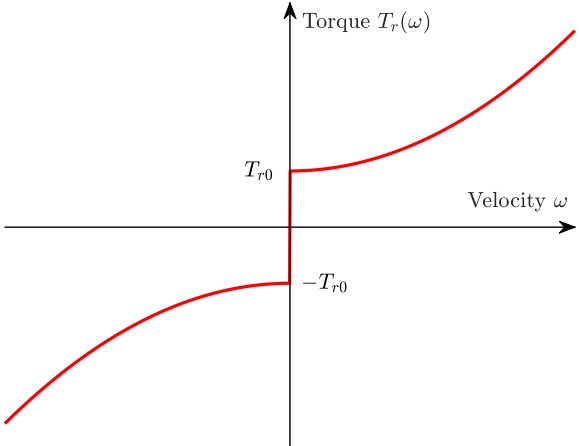

**Figure 1.** Nonlinear friction characteristic $T_r(\omega)$.

Given this model description, two alternatives seem to be promising to address nonlinear friction:

1.  Feedback disturbance compensation: In this solution, the friction term (3) is assumed as known and explicitly included in the sliding mode control design. The corresponding parameters are identified beforehand by the least-squares method. In the envisaged sliding-mode control design this would involve a time differentiation of the friction model and a compensation by means of feedback. It is clear that any changes of the friction model afterwards results in an imperfect compensation.

2.  Estimator-based disturbance compensation: In this approach, the detailed physical model for the nonlinear friction (3) is not employed at all in the control design. Instead, nonlinear friction is estimated by a Kalman filter. It turns out that the approach can be generalized by considering a lumped disturbance torque $d(t) = T_r(\omega(t)) + T_u(t)$, where $T_u(t)$ represents any further external loads torques, unmodelled dynamic effects and model uncertainty. The estimate can be subsequently used for a disturbance compensation. The modified system model is then given by

$$\frac{\mathrm{d}i(t)}{\mathrm{d}t} = \frac{1}{L}\left(u(t) - Ri(t) - K_T\omega(t)\right), \tag{4}$$

$$\frac{\mathrm{d}\omega(t)}{\mathrm{d}t} = \frac{1}{J}\left(K_T i(t) - d(t)\right). \tag{5}$$

In the sequel, the latter approach using a KF is followed because it promises a higher tracking accuracy. Moreover, the estimator dynamics can be specified appropriately in the KF design.

## 3. Control Design

The implementation of the overall control structure corresponds to the block diagram shown in Figure 2. The control input involves an equivalent control action, a disturbance compensation and a robustifying switching term. In this contribution, the switching height is adapted by means of a quasi-linear MPC.

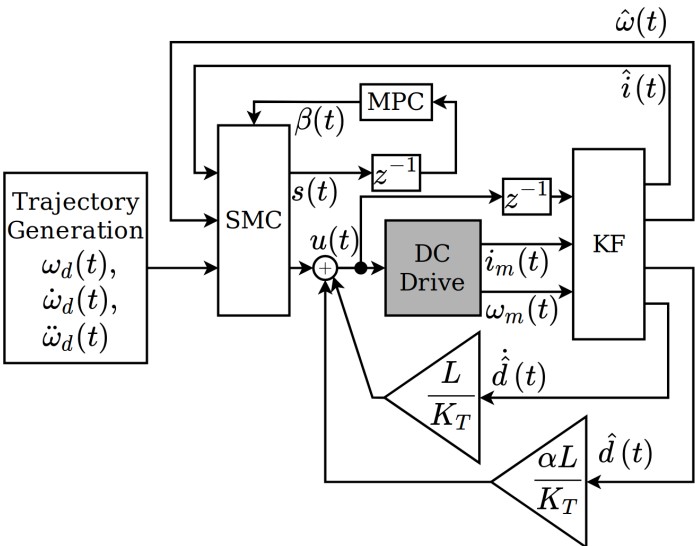

**Figure 2.** Implementation of the sliding mode control (SMC) in combination with a Kalman filter (KF) for state and parameter estimation and an model predictive control (MPC) for the adaptation of the switching height. The gray block represents the direct current (DC) drive.

### 3.1. Feedback Control Design Using SMC

Since the lumped disturbance $d(t)$ is estimated by a KF and used for a subsequent disturbance compensation, the state-space representation (4) and (5) can be used in the derivation of an integral SMC. It is worth mentioning that the estimated disturbance compensates the largest part of the lumped disturbance and significantly increases the tracking accuracy. As a result, the SMC has to cope with model imperfections that are related to the dynamics of the estimator only—leading to a significant reduction of the necessary switching height and, thereby, reducing the undesired chattering effect. The SMC design aims at a highly accurate tracking of desired trajectories for the angular velocity with smallest possible tracking errors $e(t) := \omega_d(t) - \omega(t)$. For that purpose, an integral sliding surface is introduced as follows

$$s(t) = \dot{e}(t) + \alpha e(t) + \eta \int_0^t e(\tau)\,\mathrm{d}\tau - e(0), \tag{6}$$

where $e(0)$ represents an initial error. Its presence in $s(t)$ could eliminate the reaching phase ($s(0) = 0$), see [21]. In this contribution, however, $e(0)$ is assumed to be unknown and is set to zero in the implementation. The coefficients $\alpha \in \mathbb{R}$ and $\eta \in \mathbb{R}$ have to be positive. The time derivative of the sliding surface can be easily computed and results in

$$
\begin{aligned}
\dot{s}(t) &= \ddot{\omega}_d(t) - \ddot{\omega}(t) + \alpha(\dot{\omega}_d(t) - \dot{\omega}(t)) + \eta\,(\omega_d(t) - \omega(t)) \\
&= \ddot{\omega}_d(t) - \left(\frac{K_T}{J}\frac{di(t)}{dt} - \frac{1}{J}\dot{d}(t)\right) + \alpha\left[\dot{\omega}_d(t) - \left(\frac{K_T}{J}i(t) - \frac{1}{J}d(t)\right)\right] + \eta\,(\omega_d(t) - \omega(t)) \\
&= \ddot{\omega}_d(t) - \left[\frac{K_T}{J}\left(\frac{1}{L}u(t) - \frac{R}{L}i(t) - \frac{K_T}{L}\omega(t)\right) - \frac{1}{J}\dot{d}(t)\right] + \alpha\left[\dot{\omega}_d(t) - \left(\frac{K_T}{J}i(t) - \frac{1}{J}d(t)\right)\right] \\
&\quad + \eta\,(\omega_d(t) - \omega(t)) \\
&= \ddot{\omega}_d(t) - \frac{K_T}{JL}u(t) + \frac{K_T R}{JL}i(t) + \frac{K_T^2}{JL}\omega(t) + \frac{1}{J}\dot{d}(t) + \alpha\left[\dot{\omega}_d(t) - \frac{K_T}{J}i(t) + \frac{1}{J}d(t)\right] \\
&\quad + \eta\,(\omega_d(t) - \omega(t)) .
\end{aligned}
\tag{7}
$$

It becomes obvious that the time derivative $\dot{d}(t)$ of the lumped disturbance affects the time derivative of the sliding surface $\dot{s}(t)$—and, hence, is needed in the SMC law. This explains why it is estimated

as well using a Kalman filter with a suitable disturbance model. For the derivation of the SMC law, a quadratic Lyapunov function candidate based on the integral sliding surface is considered

$$V(t) = \frac{1}{2}s(t)^2. \tag{8}$$

The time derivative of the Lyapunov function candidate can be easily calculated. It has to fulfill the sliding condition, which is chosen as follows in this paper

$$\dot{V}(t) = s(t)\dot{s}(t) \leq s(t)\left(-\lambda s(t) - \beta \text{sgn}(s(t))\right) = -\lambda s(t)^2 - \beta |s(t)|. \tag{9}$$

Now, all known terms in the time derivative $\dot{s}(t)$ are compensated for by feedback, which leads to the following expression for the equivalent control

$$u_{eq}(t) = \frac{JL}{K_T}\left[\ddot{\omega}_d(t) + \frac{K_T R}{JL}i(t) + \frac{K_T^2}{JL}\omega(t) + \alpha\left[\dot{\omega}_d(t) - \frac{K_T}{J}i(t)\right] + \eta\left(\omega_d(t) - \omega(t)\right)\right]. \tag{10}$$

Then, the time derivative of the sliding surface becomes

$$\dot{s} = -\frac{K_T}{JL}[u(t) - u_{eq}(t)] + \frac{1}{J}\dot{d}(t) + \frac{\alpha}{J}d(t). \tag{11}$$

The unknown disturbance as well as its time derivative are estimated by a Kalman filter. A disturbance compensation based on these estimates $\hat{\dot{d}}$ and $\hat{d}$, hence, contributes to a reduction of the discontinuous switching term, i.e., to a suppression of undesired chattering. The disturbance compensation law results in

$$u_{dc}(t) = \frac{L}{K_T}\hat{\dot{d}}(t) + \frac{\alpha L}{K_T}\hat{d}(t). \tag{12}$$

Finally, the switching part $u_{sw}(t)$ can be derived from the sliding condition

$$\dot{s}(t) = \frac{-K_T}{JL}u_{sw}(t) \leq -\lambda s(t) - \beta \text{sgn}(s(t)). \tag{13}$$

If the equality sign holds, the switching part is given by

$$u_{sw}(t) = \frac{JL}{K_T}\left(\lambda s(t) + \beta \text{sgn}(s(t))\right). \tag{14}$$

The overall SMC law comprises the sum of all three terms

$$u(t) = u_{eq}(t) + u_{dc}(t) + u_{sw}(t). \tag{15}$$

Outside the boundary layer – during the convergence to the sliding surface – the time derivative $\dot{s}(t)$ is governed by the nonlinear error dynamics

$$\dot{s}(t) = -\lambda s(t) - \beta \text{sgn}(s(t)). \tag{16}$$

*3.2. Adaption of the Switching Height Using MPC*

The main idea of this paper is now to use MPC techniques to determine an optimal switching height $\beta(k)$. For that purpose, the error dynamics is discretized with regard to time using the explicit

Euler method with a sampling time of $T_s = 10$ μs, and the switching height $\beta(k)$ is introduced as control input for the MPC

$$s(k+1) = (1 - \lambda T_s)\, s(k) - \beta(k) T_s \mathrm{sgn}(s(k)). \tag{17}$$

### 3.2.1. Converge Properties Outside the Boundary Layer

Outside the boundary layer, from (17), a possible discrete-time state-space representation results

$$s(k+1) = a_{k,r} s(k) + b_{k,r}\beta(k), \quad y(k) = c_k s(k), \tag{18}$$
$$\Rightarrow a_{k,r} = 1 - \lambda T_s, \ b_{k,r} = -T_s \mathrm{sgn}(s(k)), \ c_k = c = 1,$$

where subscript $r$ indicates the reaching phase characteristics. By repeated evaluations of the difference equation, the system behavior can be predicted as

$$\hat{y}(k+1) = c\, a_{k,r}\, s(k) + c\, b_{k,r}\, \beta(k) \tag{19}$$
$$\hat{y}(k+2) = c\, a_{k,r}^2\, s(k) + c\, a_{k,r}\, b_{k,r}\, \beta(k) + c\, b_{k+1,r}\, \beta(k+1),$$
$$\hat{y}(k+3) = c\, a_{k,r}^3\, s(k) + c\, a_{k,r}^2\, b_{k,r}\, \beta(k) + c\, a_{k,r}\, b_{k+1,r}\, \beta(k+1) + c\, b_{k+2,r}\, \beta(k+2),$$

etc. It is straightforward to show that the following vector expression holds

$$\hat{\mathbf{y}}(k) = \mathbf{g}_r s(k) + \mathbf{F}_{k,r}\mathbf{u}_k, \tag{20}$$

with

$$\hat{\mathbf{y}}(k) = \begin{bmatrix} \hat{y}(k+1) \\ \hat{y}(k+2) \\ ... \\ \hat{y}(k+p) \end{bmatrix}, \quad \mathbf{u}_k = \begin{bmatrix} \beta(k) \\ \beta(k+1) \\ ... \\ \beta(k+p-1) \end{bmatrix}, \tag{21}$$

and a prediction horizon of length $p$. The system matrices for use in the MPC become

$$\mathbf{g}_r = \begin{bmatrix} c\, a_{k,r} \\ c\, a_{k,r}^2 \\ ... \\ c\, a_{k,r}^p \end{bmatrix}, \quad \mathbf{F}_{k,r} = \begin{bmatrix} cb_{k,r} & \mathbf{0} & ... & \mathbf{0} \\ ca_{k,r}b_{k,r} & cb_{k+1,r} & ... & \mathbf{0} \\ ... & ... & ... & ... \\ ca_{k,r}^{p-1}b_{k,r} & ca_{k,r}^{p-2}b_{k+1,r} & ... & cb_{k+p-1,r} \end{bmatrix}, \tag{22}$$

where the prediction horizon $p$ should not be chosen as too large, considering that $b_{k,r}$ might change unpredictably. For $p = 2$, these matrices simplify to

$$\mathbf{g}_r = \begin{bmatrix} 1 - \lambda T_s \\ (1 - \lambda T_s)^2 \end{bmatrix}, \quad \mathbf{F}_{k,r} = -T_s \begin{bmatrix} \mathrm{sgn}(s(k)) & 0 \\ (1 - \lambda T_s)\,\mathrm{sgn}(s(k)) & \mathrm{sgn}(s(k+1)) \end{bmatrix}. \tag{23}$$

With given $\mathrm{sgn}(s(k))$, $\mathrm{sgn}(s(k+1))$ can be obtained from the prediction step (18) using the second element of the input vector $\mathbf{u}_{k-1}$ computed in the previous step. Given the system according to (20), an optimal input $\beta(k)$ has to be calculated that minimizes the following cost function

$$J(k) = \frac{1}{2}\left(\mathbf{y}_d(k) - \hat{\mathbf{y}}(k)\right)^{\mathrm{T}} \mathbf{Q}\left(\mathbf{y}_d(k) - \hat{\mathbf{y}}(k)\right) + \frac{1}{2}\mathbf{u}_k^{\mathrm{T}}\mathbf{R}\mathbf{u}_k, \tag{24}$$

where $\mathbf{Q} \geq 0$ and $\mathbf{R} > 0$ are symmetric non-negative definite matrices. Moreover, $\mathbf{y}_d(k)$ is the sliding surface reference trajectory for the next $p$ time steps. In this case, its elements can simply be set to zero. The corresponding solution can be stated in closed form

$$\mathbf{u}_k = (\mathbf{F}_{k,r}^T \mathbf{Q} \mathbf{F}_{k,r} + \mathbf{R})^{-1} \mathbf{F}_{k,r}^T \mathbf{Q}\left(\mathbf{y}_d(k) - \mathbf{g}y(k)\right), \tag{25}$$

where $y(k) = s(k)$ holds and the sliding mode control switching gain $\beta = \beta(k)$ is now chosen as the first element of $\mathbf{u}_k$. The MPC with the predicted sliding surface is realized with matrices (18) that do not depend on any specific systems parameters. As a result, this approach offers an intrinsic robustness regarding the prediction.

### 3.2.2. Convergence Properties Inside the Boundary Layer

As a second measure against chattering—in addition to the disturbance compensation by means of the Kalman filter—a regularized version of the switching law is employed. This leads to the definition of a boundary layer resulting from the replacement of $\mathrm{sgn}(s(k))$ by a smoothed version given by the saturation function $\mathrm{sat}\left(\frac{s(k)}{\Phi}\right)$. Inside the boundary layer, the saturation function is linear, resulting in $\mathrm{sat}\left(\frac{s(k)}{\Phi}\right) = \frac{s(k)}{\Phi}$:

$$s(k+1) = (1 - T_s \lambda)\, s(k) - T_s \frac{s(k)}{\Phi} \beta(k). \tag{26}$$

This expression contains a multiplication of a state variable with the input variable and is, hence, a nonlinear term. A first-order multivariate Taylor linearization of the function $f(\mathbf{x}_T) = \frac{s(k)}{\Phi} \beta(k)$, where the vector $\mathbf{x}_T = \begin{bmatrix} \beta(k) & s(k) \end{bmatrix}^T$ denotes the independent variables, can be performed in the operating point $\mathbf{x}_T^* = \begin{bmatrix} \beta^*(k-1) & s^*(k-1) \end{bmatrix}^T$, which contains known values at the discrete point at time $k-1$. Here, the star symbol $(\cdot)^*$ denotes the operating point and allows for representing the corresponding values in the following equations. The Taylor series expansion up to linear terms becomes

$$f(\mathbf{x}_T) \approx f(\mathbf{x}_T^*) + \nabla f(\mathbf{x}_T^*)\,(\mathbf{x}_T - \mathbf{x}_T^*), \tag{27}$$

where the operator $\nabla$ indicates the gradient of $f$, a row vector. A detailed description can be stated as follows

$$\frac{s(k)}{\Phi} \beta(k) \approx \frac{s^*(k-1)}{\Phi} \beta^*(k-1) + \frac{s^*(k-1)}{\Phi} \left(\beta(k) - \beta^*(k-1)\right) + \frac{\beta^*(k-1)}{\Phi} \left(s(k) - s^*(k-1)\right) \tag{28}$$

$$= \frac{s^*(k-1)}{\Phi} \beta^*(k-1) + \frac{s^*(k-1)}{\Phi} \beta(k) + \frac{\beta^*(k-1)}{\Phi} s(k) - 2\frac{s^*(k-1)\beta^*(k-1)}{\Phi} \tag{29}$$

$$= \frac{s^*(k-1)}{\Phi} \beta(k) + \frac{\beta^*(k-1)}{\Phi} s(k) - \frac{s^*(k-1)\beta^*(k-1)}{\Phi}. \tag{30}$$

By substituting this Taylor linearization into the difference equation, a linear first-order discrete-time model can be derived

$$s(k+1) = \underbrace{\left(1 - T_s \lambda - T_s \frac{\beta^*(k-1)}{\Phi}\right)}_{a_k} s(k) + \underbrace{\frac{-T_s s^*(k-1)}{\Phi}}_{b_k} \beta(k) + w^*(k-1), \tag{31}$$

where $w^*(k-1) = \frac{T_s}{\Phi} s^*(k-1)\beta^*(k-1)$ represents a known term consisting on past information, i.e., from the previous time step. The discrete-time system matrix and the input vector become scalars and are given by $a_k$ and $b_k$, respectively. Note that these terms are time-dependent. The output equation is scalar as well and can be stated as

$$y(k) = c_k\, s(k),\; c_k = c = 1. \tag{32}$$

In analogy to the approach outside the boundary layer, the design of a quasi-linear MPC is presented that is based on the discrete-time model above. The moving prediction horizon comprises two steps in the future. With the following vectors and matrices

$$\mathbf{g}_k = \begin{bmatrix} a_k \\ a_k \cdot a_{k+1} \end{bmatrix} = \begin{bmatrix} 1 - T_s\lambda - T_s\frac{\beta^*(k-1)}{\Phi} \\ \left(1 - T_s\lambda - T_s\frac{\beta^*(k-1)}{\Phi}\right) \cdot \left(1 - T_s\lambda - T_s\frac{\beta^*(k)}{\Phi}\right) \end{bmatrix}, \tag{33}$$

$$\mathbf{F}_k = \begin{bmatrix} c \cdot b_k & 0 \\ c \cdot a_k \cdot b_k & c \cdot b_{k+1} \end{bmatrix} = -\frac{T_s}{\Phi} \begin{bmatrix} s^*(k-1) & 0 \\ \left(1 - T_s\lambda - T_s\frac{\beta^*(k-1)}{\Phi}\right) \cdot s^*(k-1) & s^*(k) \end{bmatrix}, \tag{34}$$

$$\mathbf{w}_k = \begin{bmatrix} c \\ c \cdot a_k + c \end{bmatrix} = \begin{bmatrix} 1 \\ 2 - T_s\lambda - T_s\frac{\beta^*(k-1)}{\Phi} \end{bmatrix}, \tag{35}$$

the solution can be determined in a closed-form expression as follows

$$\mathbf{u}_k = (\mathbf{F}_k^\mathrm{T}\mathbf{Q}\mathbf{F}_k + \mathbf{R})^{-1}\mathbf{F}_k^\mathrm{T}\mathbf{Q}\left[\mathbf{y}_d(k) - \mathbf{g}_k \cdot y(k) - \mathbf{w}_k \cdot w^*(k-1)\right]. \tag{36}$$

Here, the output $y(k) = s(k)$ is identical to the current value of the sliding surface, whereas the time-varying SMC switching gain $\beta$ is chosen again as the first element of the computed input vector $\mathbf{u}_k = \begin{bmatrix} \tilde{\beta}_k(k) & \tilde{\beta}_k(k+1) \end{bmatrix}^T$. As the input $\beta^*(k)$, which shows up in $\mathbf{g}_k$ and represents the linearization point for $\beta$, is not yet available, it is substituted by the second element of the input $\mathbf{u}_{k-1}$, which corresponds to previous time step.

### 3.3. KF for the Estimation of a Lumped Disturbance Torque

In the sequel, the combined estimation of the state variables and the external disturbance as well as its time derivative is described. The design of a corresponding KF is based on the modified system model, including a double-integrator disturbance model

$$\frac{\mathrm{d}i(t)}{\mathrm{d}t} = \frac{1}{L}\left(u(t) - Ri(t) - K_T\omega(t)\right) \tag{37}$$

$$\frac{\mathrm{d}\omega(t)}{\mathrm{d}t} = \frac{1}{J}\left(K_Ti(t) - d(t))\right) \tag{38}$$

$$\frac{\mathrm{d}d(t)}{\mathrm{d}t} = \dot{d}(t) \tag{39}$$

$$\frac{\mathrm{d}\dot{d}(t)}{\mathrm{d}t} = 0 \tag{40}$$

and aims at providing estimates for both state variables, an estimate $\hat{d}$ for the unknown lumped disturbance and its derivative $\hat{\dot{d}}$. It is worth mentioning that the chain of two integrators has no input so far. Nevertheless, this integrator chain is driven by the output error feedback as well as the process noise—the stochastic part—in the framework of the KF design. As a result, the estimator states vary during the operation of the KF and highly accurate estimates are obtained for a subsequent compensation in the control structure.

Whenever feedback control is applied, it is necessary to measure selected system outputs. Under realistic conditions, however, measurements are affected by errors like deterministic offsets and stochastic disturbances, e.g., white noise processes. In such cases, a KF can be advantageously employed and provides estimates with minimum covariances. The optimality conditions include an accurate system model and the knowledge about the noise characteristics. In the given case, the system model (4) and (5)—that contains a perfectly-known part and the unknown lumped disturbance—is both complete and correct. As confirmed by the simulation results, the quality of the estimates is high and, hence, an accurate system model is obtained due to the estimates. The typical design of a KF addresses uncorrelated process noise and measurement noise that are assumed to be Gaussian,

white and with a vanishing mean value. Despite the fact that in practice the stochastic noise processes are often not perfectly known, the KF algorithm is usually still capable of providing meaningful state and disturbance estimates. The covariances can then be considered as tuning parameters like in the linear-quadratic regulator (LQR) control design. The model defining the KF prediction step can be stated in state-space form, with the input variable $u(t)$,

$$\mathbf{x}_{KF}(t) = \left[ \begin{bmatrix} i(t) & \omega(t) & d(t) & \dot{d}(t) \end{bmatrix} \right]^T, \tag{41}$$

$$\dot{\mathbf{x}}_{KF}(t) = \mathbf{A}_{KF}\mathbf{x}_{KF}(t) + \mathbf{b}_{KF}u(t), \tag{42}$$

$$\mathbf{y}_m(t) = \mathbf{C}_{KF}\mathbf{x}_{KF}(t) \tag{43}$$

with matrices

$$\mathbf{A}_{KF} = \begin{bmatrix} -\frac{R}{L} & -\frac{K}{L} & 0 & 0 \\ \frac{K}{J} & 0 & -\frac{1}{J} & 0 \\ 0 & 0 & 0 & 1 \\ 0 & 0 & 0 & 0 \end{bmatrix}, \quad \mathbf{b}_{KF} = \begin{bmatrix} \frac{1}{L} \\ 0 \\ 0 \\ 0 \end{bmatrix}, \quad \mathbf{C}_{KF} = \begin{bmatrix} 1 & 0 & 0 & 0 \\ 0 & 1 & 0 & 0 \end{bmatrix}. \tag{44}$$

The discrete-time state space model can be obtained using explicit Euler discretization with step width $T_s$, after which the random variables $\mathbf{w}_{KF}$ and $\mathbf{v}_{KF}$ are introduced to represent white, uncorrelated process and measurement noise, respectively, with normal probability distributions [22].

$$\mathbf{x}_{KF}(k+1) = \mathbf{A}_{KFd}\mathbf{x}_{KF}(k) + \mathbf{b}_{KFd}u(k) + \mathbf{w}_{KF}(k), \tag{45}$$

$$\mathbf{y}_{KF}(k) = \mathbf{C}_{KF}\mathbf{x}_{KF}(k) + \mathbf{v}_{KF}(k), \tag{46}$$

$$\mathbf{A}_{KFd} = \mathbf{I}_{4\times4} + T_s\mathbf{A}_{KF}, \tag{47}$$

$$\mathbf{b}_{KFd} = T_s\mathbf{b}_{KF}. \tag{48}$$

Based on this, the a-priori estimates are calculated in the prediction step of the Kalman filter algorithm according to

$$\hat{\mathbf{x}}_{KF}^-(k+1) = \mathbf{A}_{KFd}\hat{\mathbf{x}}_{KF}^+(k) + \mathbf{b}_{KFd}u(k). \tag{49}$$

For the first step, initial values $\hat{\mathbf{x}}_{KF}^+(0)$ can be either specified by the user or simply set to zero. The same applies to the initial uncertainty $\mathbf{P}^+(0)$ in the following equation. The a-priori estimate of the covariance matrix is

$$\mathbf{P}^-(k+1) = \mathbf{A}_{KFd}\mathbf{P}^+(k)\mathbf{A}_{KFd}^T + \mathbf{Q}_{KF}, \tag{50}$$

where $\mathbf{Q}_{KF}$ represents a $\mathbb{R}^{4\times4}$ matrix quantifying the covariance matrix of the process noise $\mathbf{w}_{KF}$,

$$\mathbf{Q}_{KF}(k) = \mathrm{E}\left(\mathbf{w}_{KF}(k)\mathbf{w}_{KF}(k)^T\right) = \begin{bmatrix} q_i & 0 & 0 & 0 \\ 0 & q_\omega & 0 & 0 \\ 0 & 0 & q_d & 0 \\ 0 & 0 & 0 & q_{\dot{d}} \end{bmatrix}. \tag{51}$$

Here, $\mathbf{Q}_{KF}$ is assumed to be constant, diagonal and positive and is treated as a tuning parameter matrix. Parameter $q_d$ is set to zero since Equation (39) is a certain relationship, while $q_{\dot{d}}$ is assigned a large value since Equation (40) is not. Parameters $q_i$ and $q_\omega$ reflect the modeling uncertainty concerning Equations (38) and (37). The Kalman gain can now be calculated as

$$\mathbf{K}(k+1) = \mathbf{P}^-(k+1)\mathbf{C}_{KF}^T(\mathbf{C}_{KF}\mathbf{P}^-(k+1)\mathbf{C}_{KF}^T + \mathbf{R}_{KF})^{-1}, \tag{52}$$

with the measurement matrix $\mathbf{C}_{KF}$ according to the measured outputs $\mathbf{y}_m(t) = \mathbf{Cx}_{KF}(t)$. Here, the measurement covariance matrix $\mathbf{R}_{KF}$ is related to the measurement noise $\mathbf{v}_{KF}$ and has properties similar to $\mathbf{Q}_{KF}$

$$\mathbf{R}_{KF} = \mathrm{E}\left(\mathbf{v}_{KF}(k)\mathbf{v}_{KF}(k)^T\right) = \begin{bmatrix} r_i & 0 \\ 0 & r_\omega \end{bmatrix}. \tag{53}$$

In the correction step of the Kalman filter algorithm, the a-posteriori estimates for covariance and states are calculated as follows

$$\mathbf{P}^+(k+1) = \left(\mathrm{I}_{4\times 4} - \mathbf{K}(k+1)\mathbf{C}_{KF}\right)\mathbf{P}^-(k+1), \tag{54}$$

$$\hat{\mathbf{x}}_{KF}^+(k+1) = \hat{\mathbf{x}}_{KF}^-(k+1) + \Delta\mathbf{x}_{KF}(k+1), \tag{55}$$

where the correction $\Delta\mathbf{x}_{KF}(k+1)$ consists of the innovation term weighted by the Kalman gain $\mathbf{K}(k+1)$

$$\Delta\mathbf{x}_{KF}(k+1) = \mathbf{K}(k+1)\left(\mathbf{y}_m(k+1) - \mathbf{C}_{KF}\hat{\mathbf{x}}_{KF}^-(k+1)\right), \tag{56}$$

with the measured current and velocity $\mathbf{y}_m = \begin{bmatrix} i_m & \omega_m \end{bmatrix}^T$. The estimated states as well as the estimated disturbance and its time derivative are used in the SMC control law.

## 4. Simulations

In this section, simulation results are presented. They were obtained using a sampling time of $T_s = 10\,\mu s$. The velocity profile to be tracked was generated from step-like signals using a second-order low-pass command-shaping filter

$$G_{LP}(s) = \frac{1}{\frac{1}{10^2}s^2 + \frac{2}{10}s + 1}. \tag{57}$$

### 4.1. Simulation Settings and Scenarios

The covariance matrices of the Kalman filter were chosen according to

$$\mathbf{R}_{KF} = \mathrm{diag}\left(0.001, 500\right), \quad \mathbf{Q}_{KF} = \mathrm{diag}\left(0.001, 0.001, 0, 0.5\right), \tag{58}$$

where the third and fourth element on the diagonal of $\mathbf{Q}_{KF}$ relate to the double integrator disturbance model, and with the initial conditions

$$\mathbf{P}^+(0) = \mathrm{diag}\left(10^3, 10^3, 0, 10^3\right), \quad \hat{\mathbf{x}}_{KF}^+(0) = \begin{bmatrix} 0 & 0 & 0 & 0 \end{bmatrix}^T. \tag{59}$$

The proposed control strategy, i.e., an adaption of the SMC switching height $\beta$ by means of MPC, is compared with two other, more classical variants. These three variants are as follows:

1. SMC with $\beta = $ const. and switching control law (14),
2. SMC with $\beta = $ const. and $u_{sw} = \frac{JL}{K_T}\left(\lambda s(t) + \beta \mathrm{sat}(s(t)/\Phi)\right)$,
3. SMC with adaptive $\beta = \beta(k)$ and $u_{sw} = \frac{JL}{K_T}\left(\lambda s(t) + \beta(k)\mathrm{sat}(s(t)/\Phi)\right)$, i.e., the proposed strategy.

The SMC design parameter $\lambda$ affecting the linear term in the sliding condition was set to zero in all variants. This corresponds to the classical choice of the sliding condition. To allow for a fair comparison, the switching height $\beta = $ const. $= 2 \times 10^7$ and the boundary layer thickness $\Phi = 200$ were iteratively tuned to achieve good tracking properties while maintaining robustness and only small chattering in the presence of disturbances. These disturbances were implemented as load torques, see Figure 3 (right).

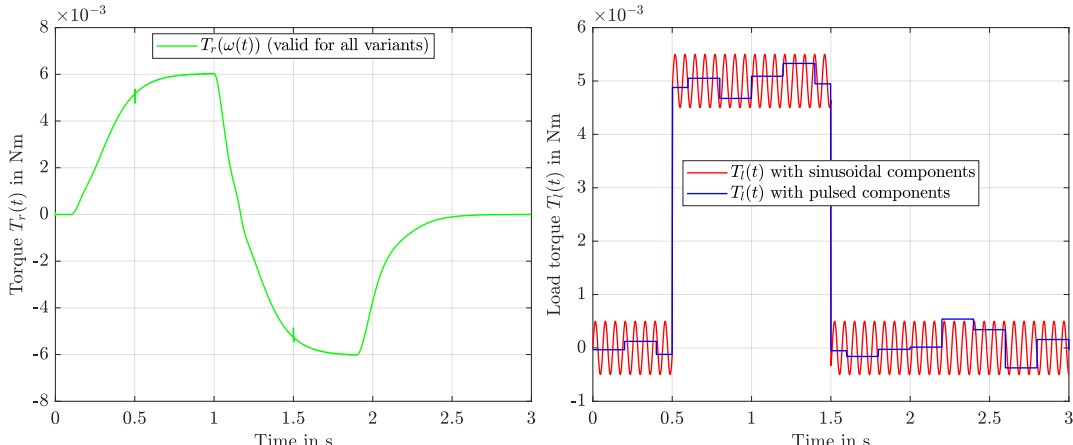

**Figure 3.** Friction torque $T_r$ (**left**) and load torque $T_l$ (**right**).

Two alternative load variants were tested: The red signal shows a sinusoidal signal with steps at 0.5 s and 1.5 s; all other figures were created using this load profile. It must be pointed out that also with pulsed load torques, i.e., the blue signal, the results were equally good. It becomes obvious that such strong discontinuities can be managed properly by an integrator disturbance model upon which the KF design is based—even though they represent the worst case for the observer part of the combined control system. Step-like changes are estimated accordingly complying with the KF estimation error dynamics.

*4.2. Results*

The resulting velocity tracking is demonstrated in Figure 4, where the tracking error and error energy are depicted, and Figure 5, which shows the angular velocity itself.

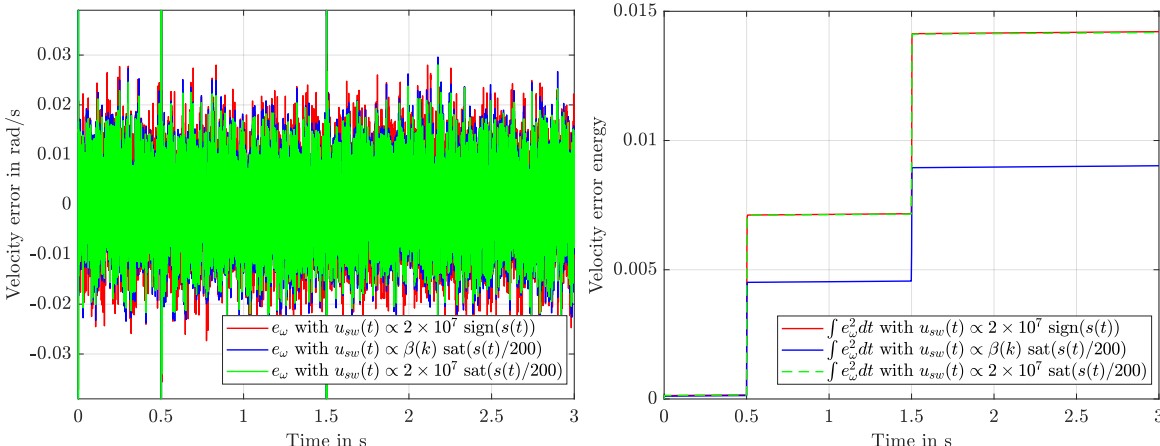

**Figure 4.** Simulated values for the tracking errors $e(t)$ for all the variants (**left**) and the integrated squared error for all the variants (**right**).

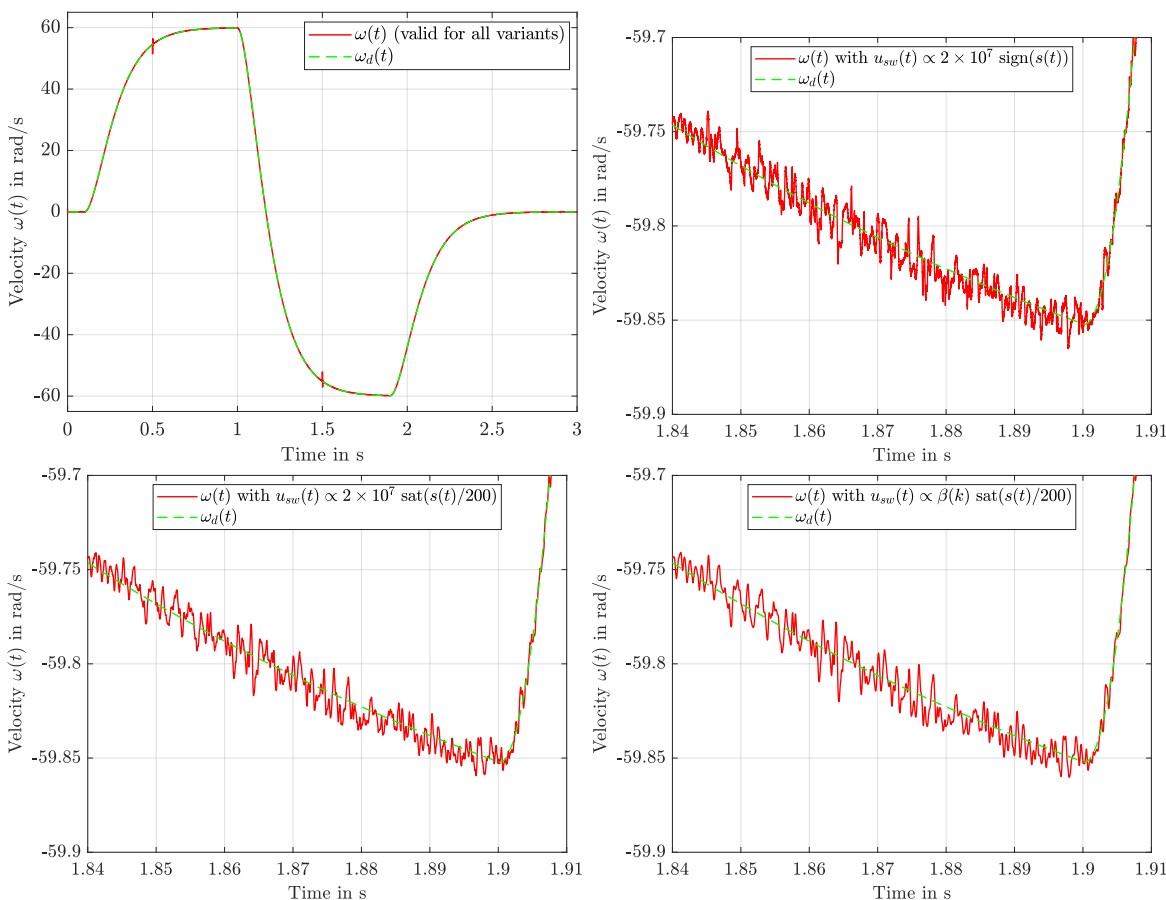

**Figure 5.** Tracking behaviour of the angular velocity $\omega$: comparison of desired and simulated values for all variants (**top**, **left**); detailed views at point with the lowest angular velocity (**top**, **right**; **bottom**).

While all the variants manage to track the desired velocity profile almost perfectly (top left of Figure 5), variant 1 shows a little more ripple (top right) than variants 2 and 3 (bottom). Additionally, variant 3 shows smaller deviations in the case of load torque steps (Figure 6), thanks to a momentarily larger switching height $\beta$. This effect results in a significantly smaller error energy, see Figure 4 (right).

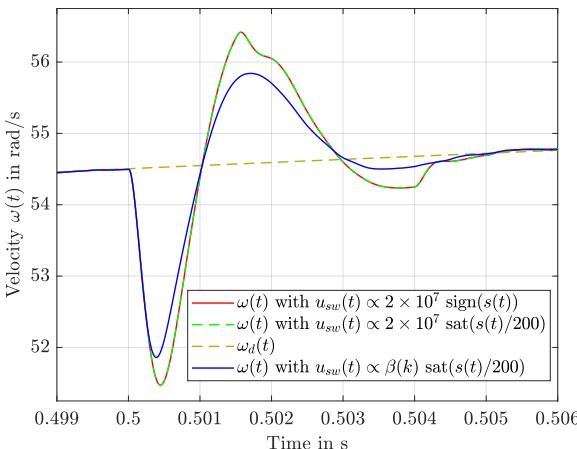

**Figure 6.** Detailed view of the velocity tracking after load torque steps.

The value of $\beta$ can be seen in Figure 7, where spikes at 0.5 s and 1.5 s are the cause for the phenomenon mentioned above. The rest of the time, $\beta$ takes relatively small values but shows a high variability.

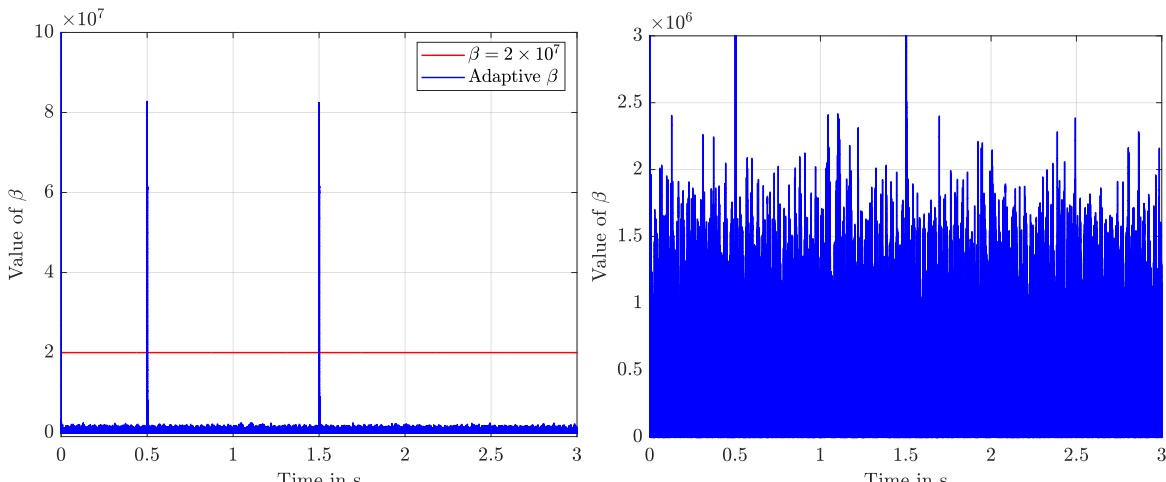

**Figure 7.** Simulated values for the switching height $\beta$ (**left**) and a detailed view (**right**).

Thanks to a smaller switching height $\beta$, a significant reduction in control input chattering is achieved. This becomes visible in Figure 8 (bottom right vs. bottom left and top right). However, the major influence stems from the equivalent control part, Equation (10), and the disturbance compensation, Equation (12), which are shown at the top left of the figure.

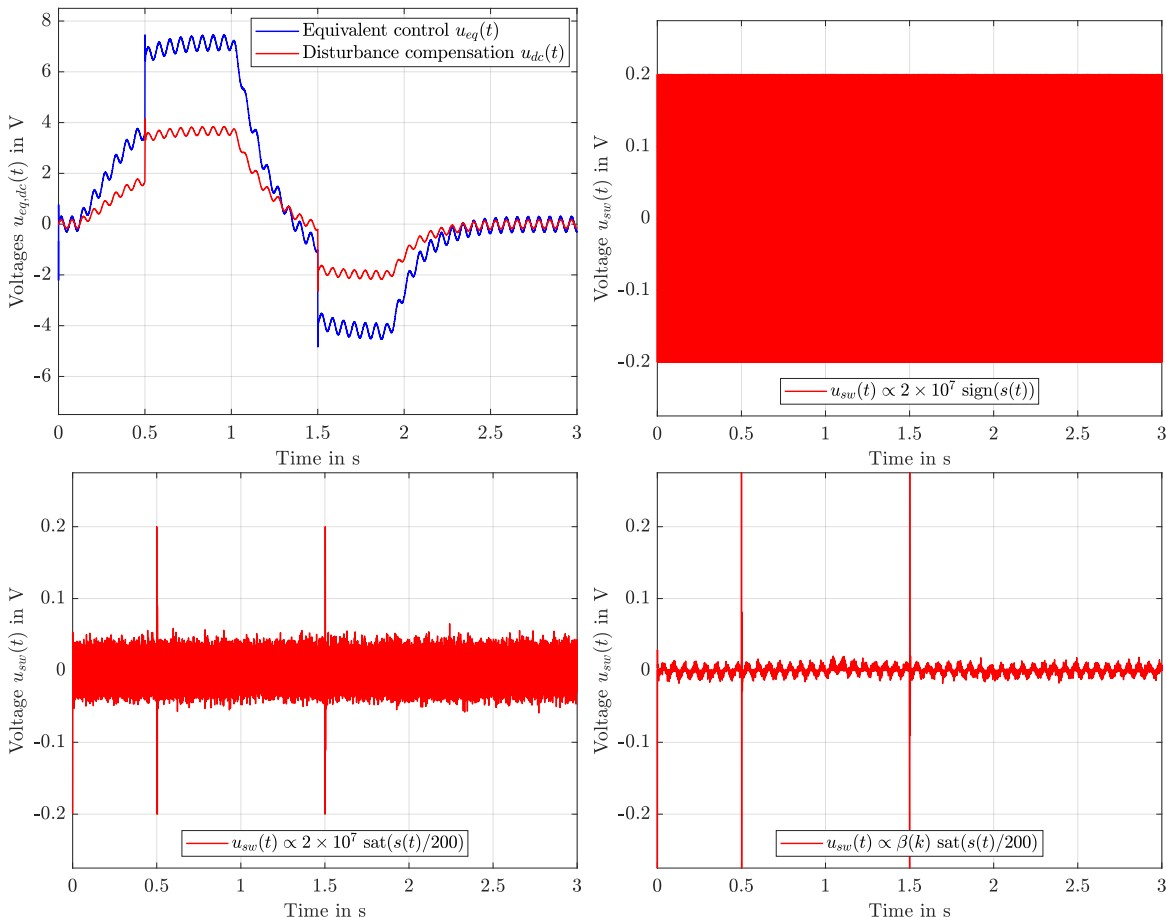

**Figure 8.** Comparison of the control inputs $u$: contribution of the equivalent control law and the disturbance compensation (**top**, **left**); detailed plots for all the variants (**top**, **right**; **bottom**).

The disturbance estimates used for compensation are provided by the KF, using only measurements of current and velocity. These estimates are shown in Figure 9. The currents resulting

from those input voltages are displayed in Figure 10 and indicate only a negligible difference between the three variants. Finally, the sliding surface $s(t)$ is depicted in Figure 11, where the influence of the external load torque becomes obvious. The SMC variants employing a boundary layer clearly achieve superior behaviour in comparison with the classical switching variant, which has a strong chattering impact in the sliding surface.

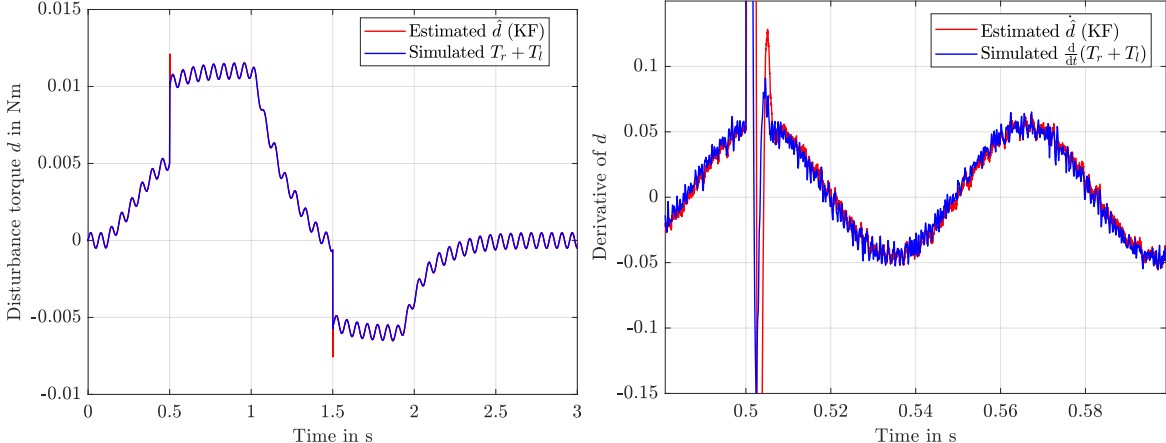

**Figure 9.** $d(t)$, $\dot{d}(t)$ and their estimates provided by the KF.

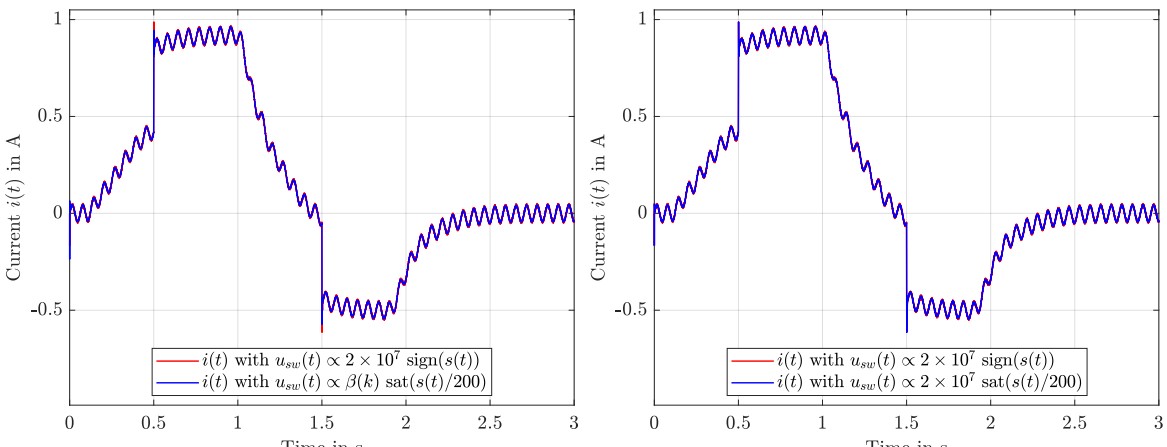

**Figure 10.** Simulated currents $i$ for all the variants.

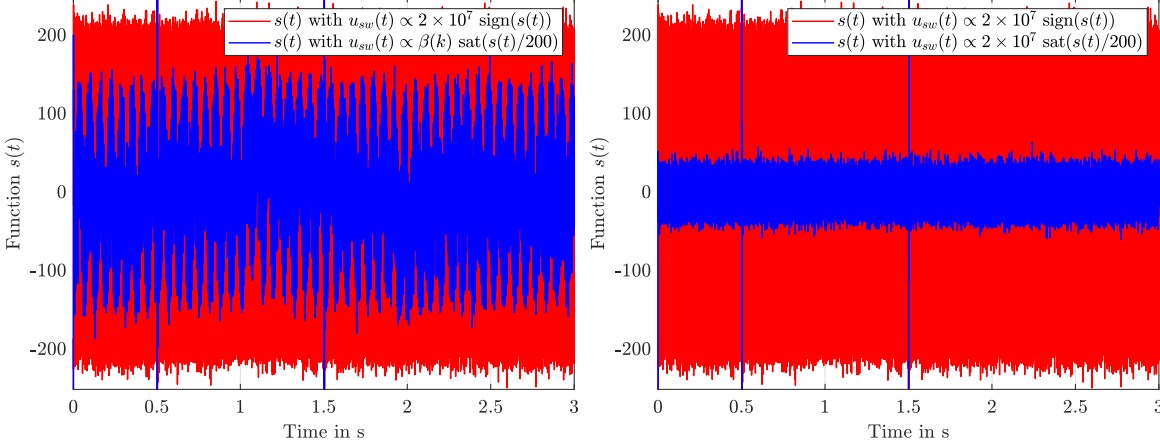

**Figure 11.** Simulated values for the sliding surface $s(t)$ for all the variants.

### 4.3. Discussion

Note that in contrast to the authors' previous work [20], upon which this contribution is based, the sliding function $s(t)$ remains in the close vicinity of zero. This positive effect can be attributed to the employed disturbance compensation provided by the KF. Some slight but negligible influence of the load torque in $s(t)$ becomes visible, which is the price of a significantly reduced chattering. In [20], due to the violation of the SMC sliding condition mentioned there, $s(t)$ was partially non-zero in the presence of large disturbances, with no tendency to converge to zero, —despite the fact that a nearly perfect velocity tracking was achieved. This phenomenon can be explained by a model mismatch concerning the dynamics of the sliding surface in closed loop, due to a missing disturbance compensation, which can yield a mismatched MPC. This problem has been solved in the given paper by means of the disturbance compensation. To further point out the benefits of the proposed approach—the combination of a disturbance estimation by a KF, finite horizon MPC and SMC gain adaptation—it is benchmarked in closed-loop simulation studies against two other widespread methods:

- A disturbance observer (DOB) according to [23] and
- A time-delay estimation (TDE), see [5].

Both are implemented to provide estimates for $d(t)$ and $\dot{d}(t)$, using the same (noisy) measurements that are available to the KF as well: current $i(t)$ and velocity $\omega(t)$. The resulting estimates can be compared in Figures 12–14. Both of these alternatives lead to similar but slightly increased error energies (the unit is omitted here) in comparison with the KF approach: 0.009024 (KF), 0.009076 (TDE) and 0.009383 (DOB). Despite a negligible difference in tracking performance, the use of either DOB or TDE in closed-loop control shows significantly larger chattering amplitudes as compared to the KF variant: $u_{sw}$ is in the range of $\pm 0.04$ V for DOB and TDE, where a range of $\pm 0.02$ V holds for the KF (apart from spikes of about $\pm 1$ V at 0.5 s and 1.5 s, respectively). Since TDE is based on the feedback of the acceleration $\dot{\omega}$, which needs to be determined via numerical differentiation of a noisy measurement signal, the performance degradation as compared to the KF becomes evident.

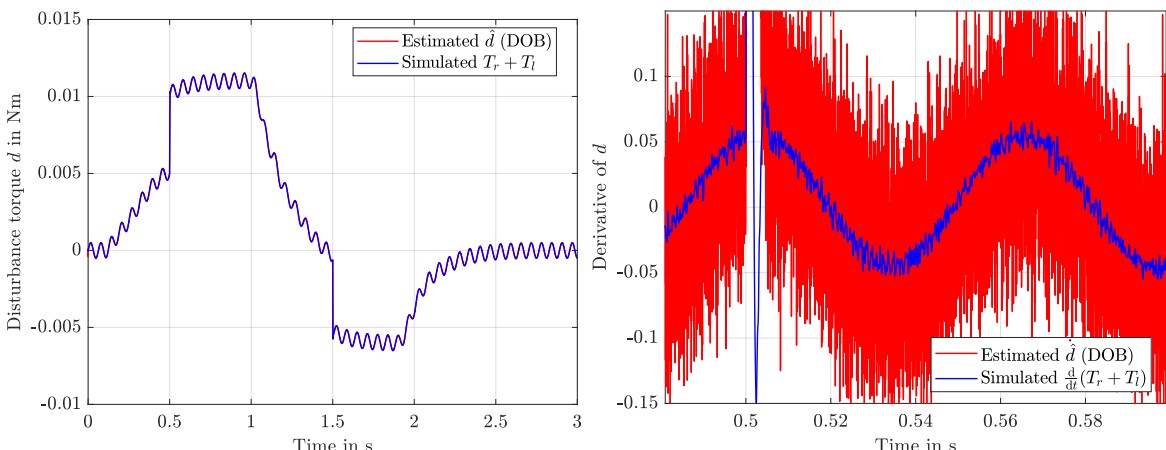

**Figure 12.** Disturbance estimates for $d$ and $\dot{d}$ using disturbance observer (DOB).

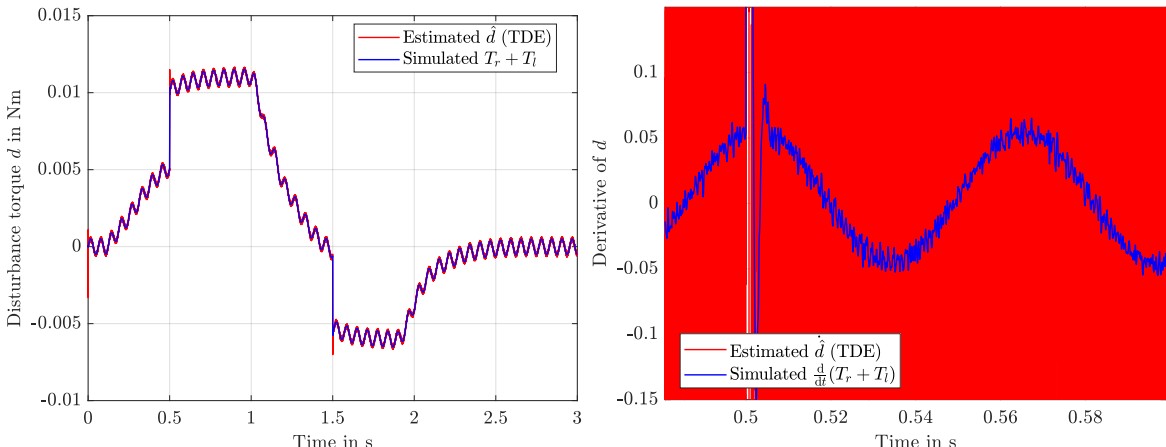

**Figure 13.** Disturbance estimates for $d$ and $\dot{d}$ using time-delay estimation (TDE).

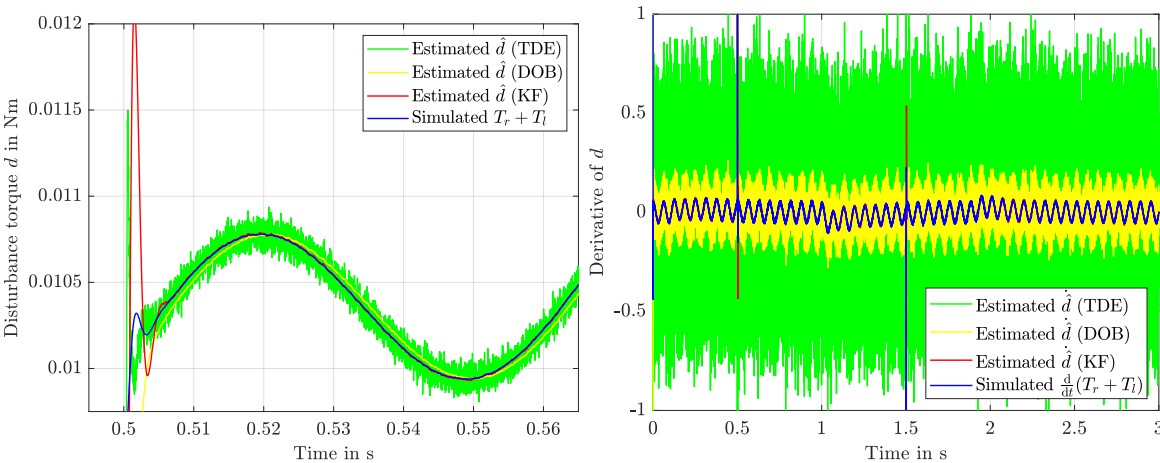

**Figure 14.** Disturbance estimates for $d$ and $\dot{d}$ using KF, DOB and TDE in comparison.

In fact, to achieve reasonably useful results, the necessary numerical derivative of the velocity $\omega(t)$ had to be low-pass filtered using a cut-off frequency of 5000 rad/s. Lower frequencies resulted in considerably worse tracking, while higher corner frequencies led to even more noise than depicted in Figure 14. The KF offers a superior behaviour in the presence of noise because it explicitly considers noise processes. Furthermore, since the KF constitutes a disturbance estimator with an integrated de-noising state observer, its state estimates for current and velocity are used to supply the SMC with state feedback, anyway. This renders the KF a perfect solution in cases like these, where state and disturbance estimates are needed. A formal proof, however, regarding the compliance with the SMC sliding condition as stated in Equation (9) using suitable choices of the MPC weighting matrices **Q** and **R**, still remains an open problem.

## 5. Conclusions

This contribution presents an adaptive tuning of the switching gain of an SMC that is achieved by means of an MPC scheme designed on the basis of the sliding surface error dynamics. To properly address disturbances and the impact of external load torque in the control approach, they are compensated for using estimates provided by a KF. The overall design is benchmarked in simulations considering a frequently used application of high practical relevance—a DC drive. The simulations clearly show that the MPC-based adaptation of the switching height represents an effective means of counteracting a drawback of classical SMC—chattering caused by a conservative choice of the switching height, which is unnecessarily high most of the time and should be reduced when permissible.

**Author Contributions:** Supervision, P.M. and H.A.; conceptualization, B.H. and P.M.; methodology, B.H., H.A. and P.M.; formal analysis, H.A.; software, B.H.; visualization, B.H.; writing—original draft preparation, B.H., P.M. and H.A.; writing—review and editing, B.H. and H.A.

**Funding:** This research received no external funding.

**Conflicts of Interest:** The authors declare no conflict of interest.

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
