# Peer review of "Gain Adaptation in Sliding Mode Control Using Model Predictive Control and Disturbance Compensation with Application to Actuators"

_information, doi:10.3390/info10050182_

Round 1

Reviewer 1 Report

The authors have proposed sliding mode control framework with gain adaptation for a dc motor in conjunction with MPC and kalman filter. However, I find the paper is unnecessarily complex. Consider the following comments:

1> First, Kalman filter is not a good tool for compensating lumped uncertainty. One needs accruate system model as in (47) and only a certain class disturbance can be tackled.

2> Again, when gain adaptation for SMC is the target, optimality is not the issue: the challenge is to tackle the uncertainty without any prior knowledge. In presence of uncertainty optimality can be never guaranteed specially in adaptive SMC. Only for nominal system it can be guaranteed. Therefore, I do not see any case for any requirement of MPC.

3> I think the authors may not be aware of the fact that different and growing field exist for adaptive sliding mode or adaptive-robust control which carries out the same objective without any such complicated route. The auhtors should follow the works as given below:

"Adaptive-Robust Control of Euler-Lagrange Systems With Linearly Parametrizable Uncertainty Bound"

"A New Design Methodology of Adaptive Sliding Mode Control for a Class of Nonlinear Systems with State Dependent Uncertainty Bound"

"Adaptive sliding mode control of a class of nonlinear systems with artificial delay"

"Adaptive-Robust Time-Delay Control for a Class of Uncertain Euler–Lagrange Systems"

"Sliding mode control with gain adaptation—Application to an electropneumatic actuator state" 

"New methodologies for adaptive sliding mode control"

"Adaptive sliding mode control with application to super-twist algorithm: Equivalent control method "

"Adaptive continuous twisting algorithm"

4> Further, MPC and KF are in discrete-time, the SMC and plant/system are in continuous time. No stability analysis is given for such case.

Overall, the authors need to study the state-of-the-art for adaptive SMC and follow the route in their work.

Author Response

Please see the attached .pdf file.

Reviewer 2 Report

The manuscript presents an adaptive tuning of the switching gain of an SMC that is achieved by means of an MPC scheme designed on the basis of the sliding surface error dynamics. To properly address disturbances and the impact of external load torque in the control approach, they are compensated for using estimates provided by a KF. The overall design is benchmarked in simulations considering a frequently used application of high practical relevance – a DC drive.

The paper is well-organized, and well-written. The following can be considered to enhance the paper;

1) The paper needs a contribution section at the end of the introduction.

2) The authors should use more recent papers to improve the readability of the paper.

3) The simulation results can be improved by adding more interpretations.

Author Response

Please see the attached .pdf file.

Round 2

Reviewer 1 Report

The paper is fine from application point of view, not at all from theoretical point of view.

The auhtors asked me a few questions. Let me try to claify as far as my knowledge:

1> In support of their argument for KF based disturbance estimator the authors mention [1]-[2] in the response. First these are two decades old paper. Furhter, the authors invited to show where it may be inapplicable. The equation (40) onwards, while deriving KF, the authors took time derivative of disturbance 'd'. I request the authors to go back to equation (3) which constitute the disturbance: it has signum function. How can one take time derivative of 'd' then? this clearly states that non-differentiable disturbance like saturation, Coulomb friction cannot be handled. I hope I have given the authors a fair answer taken from their own example.

2> The KF filter (45)-(46) relies on exact knowledge on dynamics parameters: if that is required what is the point of having adaptive SMC. Try and understand "New methodologies for adaptive sliding mode control", where the first method does not require any such thing. 

3> The authors say "..instead of rigorously proving the stability of the complete control structure. This is a common and widely accepted approach in research" and they cite [16]. Please note the difference between discontinuous control and discrte-time control. These are different. [16] nowhere uses or mixes dicrete-time control with continuous time system. Such things requires sampled data analysis. While the present work uses discrete-time model (ignoring that disturbance may not be discretized), which is a sudden jump from initial continuous SMC formulation. Stability for such thing is not guaranteed. 

4> Quoting from review response "...In practice, however, disturbances are often unmatched..."---this is not true. One can find many robotic applications which are fully actuated and always matched: a fully actuated electromechanical system is always matched.

5> again quoting "...on a sufficiently large SMC switching gain \beta, asymptotic stability can be linked to the weighting coefficient matrices..."...one employs adaptive SMC to avoid large gain not to invite.

6> quoting again "...Assuming that the usual optimality conditions are met,.." if it is assumed then it does not require analysis.

7>again. "... that a proper problem setting like the one in the given paper is suitable for quite general estimation tasks.." the considered system is not at all general.

8> "...In the general approach towards designing augmented linear Kalman filters, the unique hypothesis is that the disturbance should be modelled by an appropriate ordinary differential equation:..." this means that knowledge of structure or nature of disturbance is needed a priori. So, it cannot be completely unknown. However, the works mentioned during the first review can tackle completely unknown disturbance. For example, if one takes a model of static friction and then it turns out to be a 'Stribeck friction', then the proposed design will fail.

Author Response

Please see attached PDF file
